# TRPM4 Participates in Irradiation-Induced Aortic Valve Remodeling in Mice

**DOI:** 10.3390/cancers14184477

**Published:** 2022-09-15

**Authors:** Harlyne Mpweme Bangando, Christophe Simard, Margaux Aize, Alexandre Lebrun, Alain Manrique, Romain Guinamard

**Affiliations:** Normandie Université, UR 4650, Physiopathologie et Stratégies d′Imagerie du Remodelage Cardiovasculaire, GIP Cyceron, CHU de Caen, UNICAEN, Campus Jules Horowitz, BP 5229, 14074 Caen, France

**Keywords:** TRPM4, aortic valve, stenosis, fibrosis

## Abstract

**Simple Summary:**

Despite its benefit in cancer treatment, thoracic irradiation can induce aortic valve stenosis with fibrosis and calcification. The TRPM4 cation channel is known to participate in cellular remodeling including the transition of cardiac fibroblasts to myofibroblasts, similar to that observed during aortic valve stenosis. This study evaluates if TRPM4 is involved in irradiation-induced aortic valve damage. The aortic valve of mice was targeted by irradiation. Cardiac echography 5 months after treatment revealed an increase in aortic jet velocity, indicating stenosis. This was not observed in non-treated animals. Histological analysis revealed an increase in valvular cusp surface associated with fibrosis which was not observed in non-treated animals. The experiments were reproduced on mice after *Trpm4* gene disruption. In these animals, irradiation did not induce valvular remodeling. It indicates that TRPM4 influences irradiation-induced aortic valve damage and thus could be a target to prevent such side effects of irradiation.

**Abstract:**

Thoracic radiotherapy can lead to cardiac remodeling including valvular stenosis due to fibrosis and calcification. The monovalent non-selective cation channel TRPM4 is known to be involved in calcium handling and to participate in fibroblast transition to myofibroblasts, a phenomenon observed during aortic valve stenosis. The goal of this study was to evaluate if TRPM4 is involved in irradiation-induced aortic valve damage. Four-month-old *Trpm4^+/+^* and *Trpm4^−/−^* mice received 10 Gy irradiation at the aortic valve. Cardiac parameters were evaluated by echography until 5 months post-irradiation, then hearts were collected for morphological and histological assessments. At the onset of the protocol, *Trpm4^+/+^* and *Trpm4^−/−^* mice exhibited similar maximal aortic valve jet velocity and mean pressure gradient. Five months after irradiation, *Trpm4^+/+^* mice exhibited a significant increase in those parameters, compared to the untreated animals while no variation was detected in *Trpm4^−/−^* mice. Morphological analysis revealed that irradiated *Trpm4^+/+^* mice exhibited a 53% significant increase in the aortic valve cusp surface while no significant variation was observed in *Trpm4^−/−^* animals. Collagen staining revealed aortic valve fibrosis in irradiated *Trpm4^+/+^* mice but not in irradiated *Trpm4^−/−^* animals. It indicates that TRPM4 influences irradiation-induced valvular remodeling.

## 1. Introduction

While cancer therapies significantly improve patient survival, the appearance of late side effects may impair long-term quality of life in cancer survivors. Among these, irradiation therapies are known to favor long-term heart diseases, mostly in patients with Hodgkin disease, and breast or lung cancers. Indeed, the heart can be exposed to ionizing radiation when located, even partially, in the irradiation field [1]. Among the wide spectrum of radiotherapy-induced heart disease, cardiac valve remodeling may affect 2–37% of patients treated with radiotherapy for Hodkin’s lymphoma and 0.5–4.2% of patients treated for breast cancer [2]. The incidence of valve lesions is related to the radiation dose [3], and especially involves the aortic valve due to its anatomical location relatively close to the mediastinum [4].

Radiation-induced valvular remodeling includes thickening of the valvular cusps which might be accompanied by fibrosis and calcification leading to stenosis [2]. Interestingly, the incidence of valvular disease correlates much more with radiation dose delivered directly to the valve than to the mediastinal dose, which indicates that valvular remodeling is mostly due to a direct effect on valvular cells [5]. At the cell level, mechanisms linking radiotherapy to valvular remodeling remain to be determined. The primary cause of radiation-induced damage in cardiac tissue arises from endothelial cell injury which occurs minutes after radio treatment [1]. It induces release of inflammatory molecules responsible for the acute phase inflammatory response during a few days after irradiation. Later on, it leads to the activation of fibrogenic effector cells which can differentiate into myofibroblasts secreting collagen and α-smooth muscle actin [6]. The permanent activation of myofibroblasts leads to fibrosis which develops in the long term and ultimately results in cardiac dysfunction. While cardiac tissue is composed of cardiomyocytes, fibroblasts, and endothelial cells, the valvular tissue is mainly composed of valvular endothelial cells and valvular interstitial cells, the latter being fibroblasts. All these cell types were shown to be sensitive to irradiation and thus may participate in valvular damage after radiotherapy. Note that the fibroblast-to-myofibroblast transition is known to occur in aortic stenosis when quiescent valvular interstitial cells differentiate into myofibroblasts with the synthesis of collagen and α-smooth muscle actin. According to this, the actors involved in fibroblast transition might be involved in irradiation-induced valvular damage.

In a recent study, we observed that the differentiation to myofibroblasts of human atrial fibroblasts in culture was dependent on the Transient Receptor Potential Melastatin 4 (TRPM4) ion channel protein [7]. Pharmacological inhibition of TRPM4 reduced the development of these fibroblasts in culture. In addition, the expression of α-smooth muscle actin, a marker of myofibroblast differentiation, during culture of mouse atrial fibroblasts was reduced when the *Trpm4* gene was disrupted [7]. TRPM4 belongs to the family of TRP channels, highly expressed in cardiac tissues and participating in cardiac development and functions [8]. It is a non-selective monovalent cation channel (Na^+^ and K^+^ permeable) widely expressed in tissues with a large panel of physiological as well as pathological implications with an influence on Ca^2+^-homeostasis [9,10,11,12]. The implication of TRPM4 in cardiac activity was extensively investigated for its contribution to electrical activities and their disturbances [13]. Moreover, the channel was also shown to be involved in cardiac morphological remodeling in mice with an influence on ventricular hypertrophy [14,15] as well as ventricular fibrosis induced by pressure overload in mice [16]. Finally, TRPM4 was recently shown to be overexpressed at the protein level in aortic valves from patients suffering from aortic stenosis [17].

We thus hypothesized that TRPM4 might participates in aortic valve remodeling after radiotherapy according to its contribution to fibroblast differentiation. The goal of this study was to evaluate if the channel might be involved in irradiation-induced valvular remodeling. Using a model of targeted aortic valve irradiation in mice, we observed the development of aortic valve stenosis associated with fibrosis 5 months after irradiation treatments which was prevented in transgenic mice lacking TRPM4 expression.

## 2. Materials and Methods

### 2.1. Ethical Approval

Experiments were carried out in strict accordance with the European Commission Directive 2010/63/EU for animal care. They were conducted with authorization for animal experimentation by the French “ministère de l’enseignement supérieur de la recherche et de l’innovation” referal APAFIS#23869-2020012911013146 v6.

### 2.2. Animal Model and Irradiation

Knockout mice (*Trpm4^−/−^*) and littermate controls (*Trpm4^+/+^*) were obtained from a C57BL/6J strain as described previously [18]. Four-month-old males from both genotypes were used and raised for 5 additional months. The genotype for each animal was confirmed by genomic PCR performed on tail DNA with primers specific to wild-type and null alleles according to previously reported proceedings [18].

Animals were housed in cages to European standards (type IV), with pathogenic-free and controlled environments (21 ± 1 °C; humidity 60%; pressure 20–25 Pa; lights on 6:45 AM to 6:45 PM; enriched environment). Food and water were available ad libitum. Animals were housed at a maximum of 5 mice per cage. All efforts were made to minimize animal suffering and number.

A total of 70 animals were randomly allocated to the irradiation protocol or not. In the *Trpm4^+/+^* group, 20 mice were subjected to irradiation while 20 others were used as the control. In the *Trpm4^−/−^* group, 15 mice were subjected to irradiation while 15 others were used as the control. Irradiation was completed using the X-RAD225 Cx system (Precision X-ray Inc., North Brandford, CT, USA; CYCERON biomedical imaging platform, Caen, France). Prior to irradiation, mice anesthesia was induced by a 30 s shot of 5% isoflurane (Forene^®^, AbbeVie, Rungis, France) and afterwards anesthesia was maintained with 2% isoflurane in a mix of 30% O_2_, 70% N_2_O until the end of the irradiation protocol. The animals were placed in a supine position and the aortic valve was spotted using computed tomography to allow alignment to a homemade mouse cardiac atlas previously acquired. Planning of the irradiation exposure was performed using SmartPlan^®^ (Precision X-ray Inc., North Brandford, CT, USA). The irradiation consisted of 2 beams of 5 Gy each, 2 mm diameter, with angles of 45° and 315° respectively (120–130 s duration).

### 2.3. Echocardiography

Echocardiography was performed using an iE33 ultrasound system connected with a linear high-frequency L15-7io ultrasound transducer (Philips Healthcare, Best, the Netherlands). Measurements were performed on anesthetized animals (isoflurane 1.5% in 30% O_2_, 70% N_2_O) before irradiation and 5 months after irradiation. Animals were placed in a supine position, shaved on the chest using a chemical hair remover, and ultrasound gel was applied to optimize the detection of cardiac chambers. Atrial and aortic root diameters were measured using the TM-mode. Doppler measurements were used to evaluate aortic valve function according to the peak aortic jet velocity and mean transaortic pressure gradient (ΔP_mean_). The Doppler recordings were obtained at the level of the aortic root, the aortic valve clicks being used as reference points for Doppler measurements. All values were calculated as the mean of three consecutive measurements. The heart rate was obtained from M-mode recordings.

### 2.4. Magnetic Resonance Imaging

Cardiac magnetic resonance (CMR) imaging was performed in all animals, under isoflurane anesthesia (similar to the irradiation protocol), using a dedicated 7T magnet (Pharmascan 70/16, Bruker, Ettlingen, Germany). The animals were imaged in the supine position, with continuous dual respiratory and ECG monitoring. Anesthesia was adapted in order to maintain a respiratory frequency of 50–70 per minute [19]. Left and right ventricular function was assessed using a multi-slice dark blood intragate cine flash sequence acquired in small axis (field of view: 28 × 28 mm; matrix size: 128 × 128; spatial resolution: 0.219 × 0.219 mm/pixel; slice thickness: 560 μm; 20 frames/cardiac cycle; TR/TE: 73/2; flip angle: 25°). Image processing was performed using the Segment 3.2 software version 3.2, Segment Inc., San Francisco, CA, USA, (http://segment.heiberg.se).

### 2.5. Histological Analysis and Staining

At 5 months post-irradiation, animals were killed by cervical dislocation. The chest was open and the heart injected in the left ventricle (from the apex) with KCl 1 mol L^−1^ in the aim to remove the blood and produce muscular relaxation. The heart was then removed, immersed in formalin, and fixed in paraffin. A series of hearts were used for longitudinal slices while another was used for transversal slices performed using a microtome. For each heart, four successive 3 μm-thick slices were collected in the area of the aortic valve, and then 100 μm further, a second series of 4 slices was completed. Slices were placed on glass microscope slides for staining. For each heart, two slices were stained with classical Hematoxylin (Labelians, Nemours, France), Erythrosine (Labomoderne, Gennevilliers, France), Saffron (RAL Diagnostics, Martillac, France) (HES) using the apparatus Tissue TEK Prisma Plus (SAKURA Finetek, Villeneuve d’Ascq, France) to reveal cell nuclei (blue) as well as cytoplasm (pink) and collagen fibers (yellow). Two slices were stained with alizarin red (Sigma Aldrich, L’isle d’Abeau, France) and two others with Von Kossa to detect calcium deposits. Two slices were stained with Picrosirius Red to reveal collagen (Sirius Red, Labelians, Nemours, France; Picric acid, VWR international, Rosny sous Bois, France). In the series with transversal slices in addition to the staining indicated above, two slices for each heart were labeled using Masson trichrome to reveal extracellular matrix (hematoxylin, Labelians; fuschin acid, RAL Diagnostics; ponceau xylidin dye, RAL Diagnostics; phosphomolybdic acid, Honeywell Fluka, Illkirch, France; light green, RAL Diagnostics).

After staining, slices were digitized in white light using an Olympus VS120 microscope (Rungis, France) then analyzed using the QuPath software version 0.3.2, University of Edinburgh, Edinburgh, UK [20]. Cusp surface and sinus parameters were evaluated with manual measurements using Image J software (US NIH, Bethesda, MD, USA). For each animal, the surface of a cusp was determined as the mean of two cusps to minimize variations due to slicing level.

For Sirius red staining intensity quantification, the images were displayed in the red chromaticity mode using Qu Path, then intensity was quantified using Image J. For the longitudinal slices, for each valve, the intensity was determined as the mean of intensity of 3 rectangles of 200 × 50 μm in different areas of the cusps. Then, the mean intensity of 2 areas (200 × 50 μm) of the image outside of the tissue was subtracted to eliminate background signal. For the transversal slices, the same proceeding was applied with rectangles of 100 × 50 μm. For Masson trichrome staining quantification, the same proceeding was applied while the images were displayed in the green chromaticity mode using Qu Path and measured rectangles were 100 × 50 μm. Finally for Sirius red as well as Masson trichrome staining intensity quantification in the atrium and ventricle, the same proceeding was applied with squares of 100 × 100 μm.

### 2.6. Data Analysis

Results are reported as mean ± S.E.M. Data were analyzed using Addinsoft Xlstat 2021 (New York, NY, USA). To compare data between groups, a non-parametric Kruskal–Wallis test was used, followed by a post-hoc Dunn’s test. When appropriate, i.e., for comparison of parameters between M0 and M5 in the same animal, a paired test was used. Statistically significant difference was achieved for values of *p* ≤ 0.05. Statistically significant differences are indicated in the graphs by asterisks. N refers to the number of mice used.

## 3. Results

Four-month-old mice from *Trpm4^+/+^* as well as *Trpm4^−/−^* genotypes were randomly attributed to two groups, one being treated by aortic valve irradiation while the other was not. At the onset of the protocol (baseline), animals’ weights were similar between all groups (Table 1). Cardiac parameters were evaluated by echocardiography and CMR imaging measurements before the irradiation protocol and 5 months after irradiation (M5). At M5, hearts were removed for histological analysis. In the following paragraphs and figures, mice treated by irradiation were designated RT+ while mice not treated by irradiation were designed RT−.

### 3.1. Irradiation-Induced Functional Remodeling

Echocardiography at baseline then M5 revealed a significant time-dependent increase in maximal aortic valve jet velocity as well as mean transaortic pressure gradient (ΔP_mean_) in all groups indicating an effect of aging. However, irradiation significantly enhanced this increase in Trpm4^+/+^ but not in Trpm4^−/−^ animals (Figure 1). At M5, maximal aortic valve jet velocity from RT+ Trpm4^+/+^ was significantly higher than from RT− Trpm4^+/+^ (240.9 ± 17.2 and 185.1 ± 7.9 cm·s^−1^, respectively) while no difference was observed in Trpm4^−/−^ depending on irradiation treatment (210.9 ± 8.3 cm·s^−1^ for RT+ and 212.3 ± 11.7 cm·s^−1^ for RT−) (Figure 2). Similar variations were observed for ΔP_mean_ at M5 with a significantly higher value for RT+ compared to RT− Trpm4^+/+^ mice (11.7 ± 2.1 and 6.5 ± 0.7 mmHg, respectively) but no significant difference between RT+ and RT− Trpm4^−^^/−^ animals (8.4 ± 0.6 mmHg and 8.3 ± 1.4 mmHg, respectively) (Figure 2 and Table 1). No aortic valve regurgitation was detected in any group (Figure 2).

### 3.2. Irradiation-Induced Morphological Remodeling

CMR imaging and echography measurements at M5 did not reveal any effect of irradiation on ventricular parameters as well as left atrial diameter and aortic root diameter, whatever the genotype (Table 1 and Table 2). Animal weights were also similar between all groups but significantly higher than that of baseline (Table 1). Note that left ventricular hypertrophy appeared in Trpm4^−/−^ mice compared to Trpm4^+/+^ animals, as already reported in this mouse strain [14,15,21], but was not affected by irradiation.

Aortic valve morphological parameters were determined using surface measurements on histological slices from hearts collected at M5. Regarding longitudinal slices, the surface of a cusp was similar in Trpm4^+/+^ and Trpm4^−/−^ animals (Figure 3) when non-irradiated. Irradiation induced a significant increase in cusp surface in Trpm4^+/+^ mice but not in Trpm4^−/−^ animals (Figure 3). The same results were observed when the total surface of the cusp and annulus (insertion of the cusp) was taken into account.

To evaluate the relation between morphological and functional irradiation-induced cusp remodeling, the maximal aortic valve jet velocity was plotted as a function of the surface of one cusp as well as the surface of the cusp and annulus (Figure 4). The Pearson’s product moment correlation revealed r coefficients of 0.61 and 0.62, which correspond to *p* < 0.001, indicating a significant correlation.

Aortic sinus remodeling was evaluated on hearts sliced in the transversal axis to reveal the entirety of the sinus (Figure 5). The sinus area (tissue area) was not significantly different between Trpm4^+/+^ and Trpm4^−/−^ animals in the absence of irradiation. Irradiation induced an increase in aortic sinus area from Trpm4^+/+^ mice, even if not reaching significance, but not in Trpm4^−/−^ animals. No variation in sinus internal diameter was induced neither by gene disruption nor irradiation (Figure 5).

### 3.3. Effect of Irradiation on Aortic Valve Fibrosis

Fibrosis was first evaluated by Sirius red staining on aortic valve cusps using longitudinal heart slices. Staining intensity in the cusp area was significantly increased by 38% in irradiated Trpm4^+/+^ mice compared to non-treated controls while no significant effect of irradiation was detected on Trpm4^−/−^ animals (Figure 6).

Fibrosis was also evaluated using Masson trichrome staining on transversal slices. Cusp as well as annulus labeling was increased in irradiated Trpm4^+/+^ mice even if not reaching significance while no effect of irradiation was observed on Trpm4^−/−^ animals (Figure 7). On the other hand, labeling of the sinus was significantly increased by 4-fold after irradiation in the Trpm4^+/+^ mice but no effect was detected in Trpm4^−/−^ animals (Figure 7).

To evaluate fibrotic remodeling on the atrium and ventricle, Sirius red staining was measured on the right atrial and upper left ventricular area using longitudinal heart slices. No significant difference was observed depending on genotype or irradiation (Figure 8). Fibrotic remodeling on the atrium and ventricle was also evaluated using Masson trichrome on transversal slices. Quantification of Masson trichrome labeling on atrial as well as ventricular sections in the vicinity of the valve did not reveal any effect of irradiation. Green intensity labeling (in arbitrary unit) was 21.8 ± 8.4 (*n* = 9) and 24.3 ± 8.9 (*n* = 10) at the atrial level for Trpm4^+/+^ RT− and RT+ animals, respectively. It was 11.1 ± 5.7 (*n* = 9) and 12.9 ± 5.2 (*n* = 10) at the ventricular level for Trpm4^+/+^ RT− and RT+ animals, respectively.

### 3.4. Effect of Irradiation on Aortic Valve Calcification

Aortic valve calcification was assessed with alizarin red staining on cusps using longitudinal heart slices. No significant red labeling was detected on cusps or annulus, regardless of genotype or irradiation (see Figure 9 for Trpm4^+/+^ mice). As a control of calcification staining, an example of stained cartilage from a bronchus is provided in Figure 9. The lack of calcification induced by gene disruption or irradiation was confirmed by Von Kossa staining (Figure 9). Note that in alizarin red as well as Von Kossa staining experiments, brown labeling was detected and attributed to melanocytes staining as previously reported in such preparation [22].

### 3.5. Tricuspid and Bicuspid Aortic Valves in Trpm4^−/−^ Mice

Aortic valve transversal slices allow to determine the number of cusps. All Trpm4^+/+^ mice (*n* = 19) exhibited tricuspid valves whereas 2 over 12 Trpm4^−/−^ mice exhibited bicuspid instead of tricuspid aortic valves. Examples of such valves are provided Figure 10.

Histological slices performed at 5 months post-treatment in the transversal axis (3 μm thickness) of the aortic valve for non-irradiated Trpm4^+/+^ mice and Trpm4^−/−^ mice. Left image shows a tricuspide aortic valve representative for Trpm4^+/+^ mice. Other images represent the heterogeneity of valves in Trpm4^−/−^ mice with either tricuspid (middle image) or bicuspid (right image) aortic valves. Slices were stained with hematoxylin and eosin (HES).

## 4. Discussion

Our data indicate that specific irradiation targeting the aortic valve area induces a morphological remodeling leading to a functional remodeling of this valve which is prevented by *Trpm4* gene disruption. The TRPM4 channel therefore appears as a participant in the process linking irradiation to aortic stenosis.

Our animal model of valvular irradiation reproduces several types of valvular damage observed in patients after radiotherapy (stenosis and fibrosis) [2]. The specific targeting of the aortic valve in our model differs from previously reported studies in mice in which either the entire heart or even animal was irradiated [23,24] or the beam was restricted to the ventricular apex [25]. This is a specificity of our study since a recent review reported that among 159 studies using models of irradiation-induced cardiac remodeling in rodents, only 4% targeted a restricted area of the heart while others undertook whole heart or whole animal irradiation [26]. Some of these studies in mice with specific irradiation of the heart revealed an increase in fibrosis detected by Masson trichrome staining, observed at 5 months at the ventricular level [23,25], which was also reported in rats [27]. Interestingly, in our study, we did not observe such fibrosis neither at the ventricular level nor at the atrial level, while it was present in valvular structures, consistent with a specific targeting of the valve. Our functional recordings using cardiac magnetic resonance imaging revealed a ventricular hypertrophy without modification of the ejection fraction in *Trpm4^−/−^* animals compared to *Trpm4^+/+^* mice, in accordance with previous observations made using echography measurements in this mouse strain [14,21]. No effect of irradiation was observed on these ventricular parameters. According to this, the functional effects that we observed on the maximal aortic valve jet velocity and mean transaortic pressure gradient most probably occur following valvular remodeling rather than ventricular remodeling.

Even if our data demonstrate that valve irradiation leads to fibrosis and that TRPM4 participates in the phenomenon, the molecular mechanisms as well as cell types affected by ionizing radiation remain to be identified. Following the acute stage of irradiation-induced cardiac remodeling characterized by endothelial dysfunction and inflammation, the chronic stage is mainly characterized by persistent fibrosis as we observed at 5 months in our model. Signaling involving transforming growth-factor (TGF-β1) which is known to stimulate transformation of cardiac fibroblasts to myofibroblasts and collagen synthesis [28] might be an interesting pathway to evaluate in the irradiation-induced fibrosis that we observed since increased TGF-β1 signaling occurs weeks after radiotherapy [29]. The classical TGF-β1 signaling pathway includes its binding to type I and II receptors inducing phosphorylation of Smad transcription factors leading to extracellular matrix gene activation [30]. Connective tissue growth factor (CTGF) activation may also participate as it is known to be a downstream mediator of the TGF-β1-induced activation of fibroblasts [30]. Evidence indicates that CTGF participates in the differentiation of cardiac fibroblasts to myofibroblasts after induction by TGF-β1 and its expression is essential for persistent fibrosis [31]. CTGF expression is known to be sensitive to Ca^2+^-homeostasis through Ca^2+^–calcineurin–NFAT pathway [32]. This might be a linker for the implication of TRPM4. Indeed, TRPM4 was shown to be indirectly involved in the regulation of the calcineurin–NFAT pathway in a model of pressure-overload-induced cardiac hypertrophy in mice [16]. In this model, the calcineurin–NFAT pathway is inhibited by CaMKII under the regulation of TRPM4. Note that the authors also reported that *Trpm4* gene disruption results in a reduction in pressure-overload-induced cardiac fibrosis [16]. Interestingly, TRPM4 expression in human ventricular fibroblasts in culture was increased after TGF-β incubation [33] and this was also observed on mouse atrial fibroblasts in culture [34]. Specific data also suggest that the TGF-β1 and CTGF pathways might be independent. Indeed, irradiation of the whole heart of C57Bl6 mice revealed Rho activation at 20 weeks after treatment followed by an increase in CTGF and extracellular matrix deposition at 40 weeks but without detectable variation of TGF-β1 [35].

Metabolism-related proteins may also be involved in irradiation-induced remodeling. To support this hypothesis, it was shown that cardiac mitochondrial respiratory capacity was reduced 40 weeks after cardiac irradiation in mice at 2 Gy [36]. Note that this was confirmed in a recent study in mice after heart irradiation at 16 Gy which induces an increase in extracellular matrix proteins and variation in metabolism-related proteins detectable after 5 months, a time similar to that used in our study [23]. Dysregulation of the metabolism status with modification of ATP levels may impact TRPM4 in two different ways. First, by modifying Ca^2+^-homeostasis which is a strong regulator of TRPM4 but also because TRPM4 is sensitive to intracellular ATP [37].

Another candidate to link irradiation, TRPM4, and fibrosis may implicate the oxidative status. Indeed, irradiation leads to reactive oxygen species generation (ROS) which is particularly observed in radiation-induced fibrosis [38]. Ionizing radiation interacts with water molecules leading to the production of ROS, in which the high level persists a long time after exposure through induction of mitochondrial dysfunction, participating in late side effects. The implication of TRPM4 in ROS level increase and/or ROS-induced damage was shown in several models. It was recently observed on an adenocarcinoma cell line LS174T that knockdown of *TRPM4* depresses ROS generation [39]. At the cardiac level, hydrogen peroxide (H_2_O_2_) increases ROS production as well as intracellular Ca^2+^ level in an H9c2 rat cardiomyocyte cell line, which was prevented by *Trpm4* gene disruption [40]. This implication of TRPM4 was also observed in endothelial cells. In particular, TRPM4 favors H_2_O_2_ enhanced migration of human umbilical vein endothelial cells (HUVECs) [41]. This effect might occur through a modulation of TRPM4 biophysical properties. Indeed, it was shown in a heterologous expression system that H_2_O_2_ increases sustained TRPM4 current by reducing its desensitization [42]. It could also occur through gene overexpression since H_2_O_2_ treatment causes an increase in TRPM4 expression in HUVECs [43]. Note that TRPM4 was detected in various types of endothelial cells including: at the mRNA level, human pulmonary arteries endothelial cells [44]; at the protein level, mouse aortic endothelial cells [45]; at the functional level, HUVEC [46] as well as mouse and human brain endothelial cells [47]. According to these results, valvular endothelial cells which cover the valvular cusps might be affected by irradiation.

In addition, valvular interstitial cells which are the main cell type in the cusps might also be affected by irradiation since they produce extracellular matrix, thus, participating in fibrosis. Data on the irradiation effect on valvular interstitial cells are sparse. Porcine valvular interstitial cells in 2D as well as 3D cultures subjected to irradiation were shown to undo a transition from the myofibroblastic to the osteoblastic phenotypes associated with calcification, and DNA damage leading to a decrease in cell viability [48]. Another study reported that irradiation of human valvular interstitial cells in culture induced an increase in osteogenic factors within 24 h [49]. Human valvular interstitial cells were shown to express a low level of TRPM4 protein [17] but no functional channel was reported yet. However, more generally, TRPM4 is known to be expressed in the fibroblast family to which valvular interstitial cells belong. In particular, it is functionally expressed in human as well as mouse atrial fibroblasts [7]. Application of 9-phenanthrol, a TRPM4 inhibitor [50], during culture of human atrial fibroblasts, reduces their growth [7]. Moreover, the transition from atrial fibroblasts to myofibroblasts was reduced in *Trpm4^−/−^* compared to *Trpm4^+/+^* mice cells in culture [7]. It indicates that TRPM4 participates in this transition which is known to occur during valvular stenosis when quiescent valvular interstitial cells differentiate into myofibroblasts enhancing extracellular matrix production. Interestingly, it was recently shown that *Trpm4* gene disruption, specifically in cardiomyocytes, results in a reduction in ventricular fibrosis induced by transverse aortic constriction in mice [16], indicating that TRPM4 also participates in cardiac fibrosis through those cells.

In addition to endothelial cells and fibroblasts, cardiomyocytes might also be involved since these cells are known to functionally express TRPM4 [13] and be sensitive to irradiation [35]. However, as described above, since we did not detected variations in functional and morphological ventricular parameters after irradiation, we hypothesized that cardiomyocytes are not the main cell actor in our model with specific valvular targeting.

Beside the effect of irradiation, a learning from our study is the absence of significant aortic valve dysfunction in *Trpm4^−/−^* mice, since these animals have cusp surface and aortic flux parameters similar to the wild type. Note that valvular morphological parameters measured for wild-type mice in our study were similar to those previously reported for animals of the same age (9 months) in another study [22]. However, it is intriguing that in our experiments where transversal slicing was performed, we detected 2 *Trpm4^−/−^* mice with apparent bicuspid aortic valve over the 12 animals used (17%) while none were detected over the 19 *Trpm4^+/+^* mice. According to the small number of animals, it does not reach significance, however, a recent review of the literature based on 1823 mice reported that a bicuspid aortic valve in mice occurs in only 0.3% of wild-type animals [51]. Even more, specifically in the studies using C57BL/6 mice similar to the strain used in our study, only 2/366 animals exhibited bicuspid aortic valves [51,52]. It thus indicates that the bicuspid valve is rare in C57BL/6 mice. This occurrence (2/366) is significantly different from that observed in *Trpm4^−/−^* mice from our study (2/12), according to the Fisher exact test. It is thus conceivable that TRPM4 participates in the development of the aortic valve even if this remains to be confirmed in a larger population. This is, to our knowledge, the first description of valvular malformation in *Trpm4^−/−^* mice.

A limitation of our study comes from the so called “long-term” effect which is, in our protocol, evaluated at 5 months while, in human, valvular dysfunctions occur within years or decades after radiation therapy in patients [1]. However, it is known that the first detectable signs of irradiation-induced fibrosis in humans begins 4–12 months after radiation therapy, then develops in the following years [38]. Our study may thus represent the initial step of valvular remodeling. It is conceivable that it could lead to calcification after additional months. In addition, a strength of our study is the evaluation of the in-vivo effect of irradiation instead of in vitro studies, allowing a better appreciation of long-term effects observed in patients treated with radiotherapy since it was reported that effects of irradiation performed in vivo are stronger than those observed in vitro on cell culture [53]. Regarding irradiation intensity, the single dose of 10 Gy that we used in our study was under the median physical dose (16 Gy) used in in vivo studies evaluating cardiac radiotoxicity in rodents [26] but might be difficult to extrapolate to doses used during radiotherapy in cancer treatment. Indeed, a discrepancy may appear between a single dose treatment that we applied and repeated treatments as observed in human medicine. Such unique dose proceeding is used in the vast majority of studies (85%) evaluating cardiac radiotoxicity in rodents [26]. Note that non-irradiated mice were not subjected to a sham irradiation in our study. However, since these mice were also subjected to anesthesia for echography and cardiac MRI, one can consider that the effect observed in irradiated mice can be attributed to irradiation rather than the anesthesia protocol. Another limitation came by the fact that our study was performed only on male mice. This choice was made since males are known to be more prone to valvular stenosis with a twofold excess risk compared to females in humans [54]. However, females are also known to develop aortic stenosis after radiation treatment, for example, in the case of breast cancers. According to this, it is probable that our data would also be observed in females even if this remains to be explored.

It is also difficult to predict what the effect of TRPM4 inhibition would be in vivo if this is used to prevent radiotherapy-induced valvular damage. Indeed, TRPM4 has a large distribution in tissues and contributes to numerous physiological processes including cardiac electrical activity [9,13]. At present, very few TRPM4 inhibitors are available and only two studies reported the effect of in vivo acute application of the TRPM4 inhibitors: one with 9-phenanthrol injected in rats which results in reduction in arterial pressure and heart rate [55]; another with meclofenamate injected in mice which results in QTc reduction on an ECG [56].

## 5. Conclusions

Our study identifies TRPM4 as an actor of the irradiation-induced aortic valve damage. However, the mechanisms and even the cell types involved in the phenomenon are unknown and must be described before elaborating any therapeutic strategies to prevent radiotherapy side effects. Regarding the implication of TRPM4 in the cardiovascular system, our study revealed an implication in aortic valve remodeling, indicating that it could be valuable to investigate its potential role in other aortic valve dysfunctions.

## Figures and Tables

**Figure 1 cancers-14-04477-f001:**
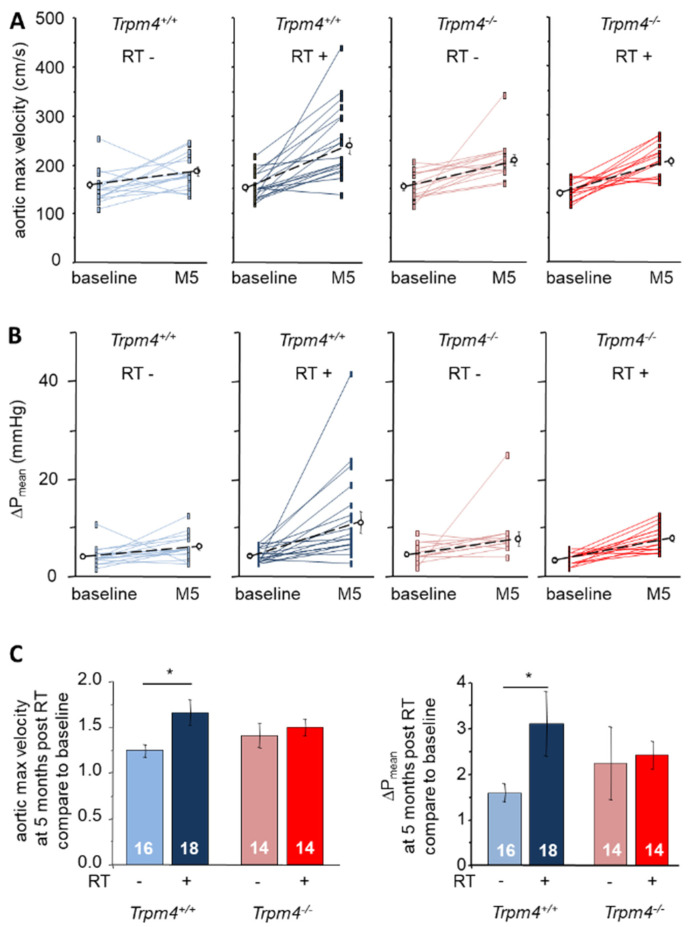
Evolution of the maximal aortic valve jet velocity and mean pressure gradient after radiation treatment. (**A**) Maximal aortic valve jet velocity measured at baseline and 5 months after irradiation treatment (RT+) or not (RT−) in *Trpm4^+/+^* and *Trpm4^−/−^* mice. Plain lines indicate paired data for the same mouse at baseline and 5 months. Dotted lines and open circles indicate the mean ± S.E.M for each group. (**B**) Mean pressure gradient (ΔPmean) measured at baseline and 5 months as explained in A. (**C**) For each mouse, maximal aortic valve jet velocity (**left**) as well as ΔP_mean_ (**right**) measured at 5 months post-treatment was reported compared to that of baseline (4-month-old mice). Histograms indicate the mean ± S.E.M. for each group. In all figures, numbers in bars indicate the number of animals and * indicates a significant difference between groups (*p* < 0.05).

**Figure 2 cancers-14-04477-f002:**
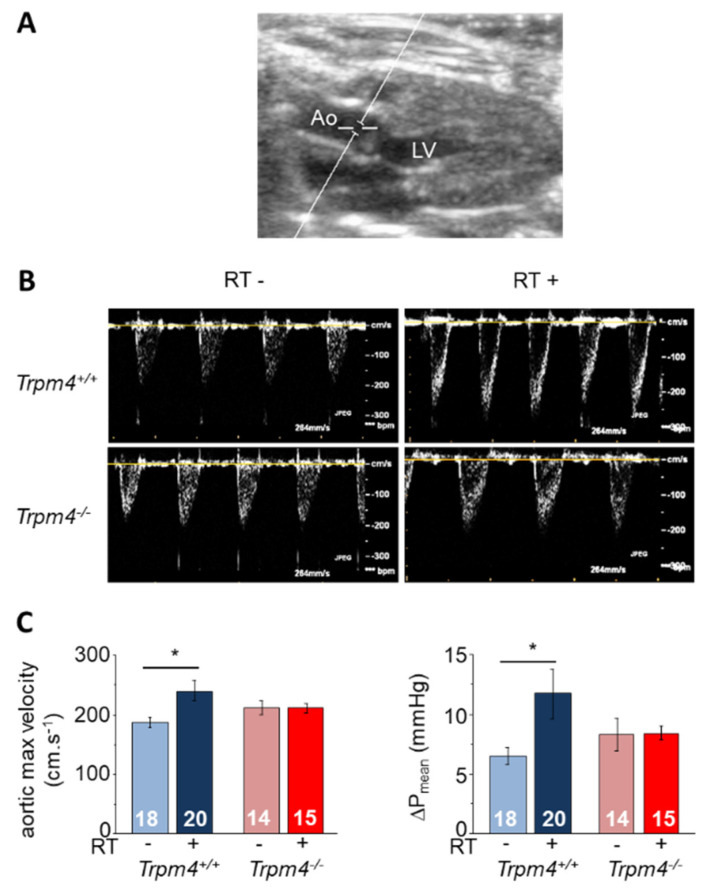
Functional valvular remodeling 5 months after radiation treatment. (**A**,**B**) Examples of transthoracic echocardiography in the Doppler mode for *Trpm4^+/+^* and *Trpm4^−/−^* without (RT−) or with (RT+) irradiation at 5 months post-treatment on anesthetized 9-month-old male mice. Measurements were made at the aortic valve level to reveal aortic flux velocity. The image provided in A allows to visualize where the measurements were taken. LV (left ventricle); Ao (aorta). (**C**) Mean maximal aortic valve jet velocity (**left**) and mean pressure gradient (ΔP_mean_) (**right**) for each group according to the echocardiography. Data are expressed as mean ± S.E.M. * indicates a significant difference between groups (*p* < 0.05).

**Figure 3 cancers-14-04477-f003:**
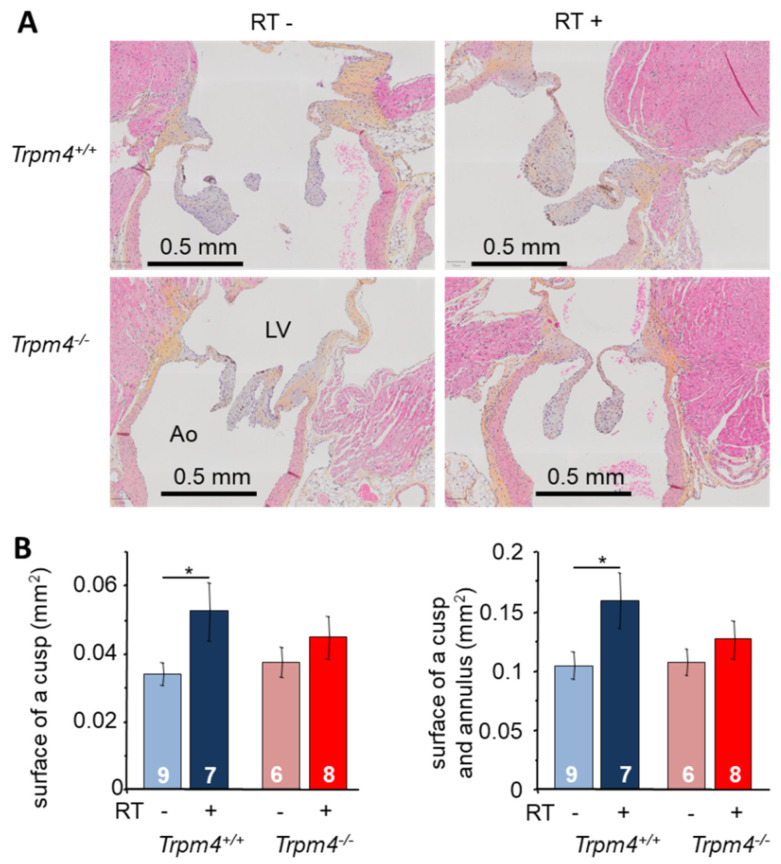
Morphological remodeling after irradiation: valvular remodeling after irradiation in the longitudinal axis. (**A**) Histological slice performed at 5 months post-treatment in the longitudinal axis (3 μm thickness) of the aortic valve representative for each group. The left ventricle (LV) is in the upper side and aorta (Ao) in the lower side. Slices were stained with hematoxylin, eosin, and saffron (HES) to reveal cell nuclei (blue) cytoplasm (pink) as well as collagen fibers (yellow). (**B**) The surface of a cusp (**left**) as well as the surface of a cusp and its annulus (**right**) were measured for each mouse. Histograms indicate the mean ± S.E.M. for each group. * indicates a significant difference between groups (*p* < 0.05).

**Figure 4 cancers-14-04477-f004:**
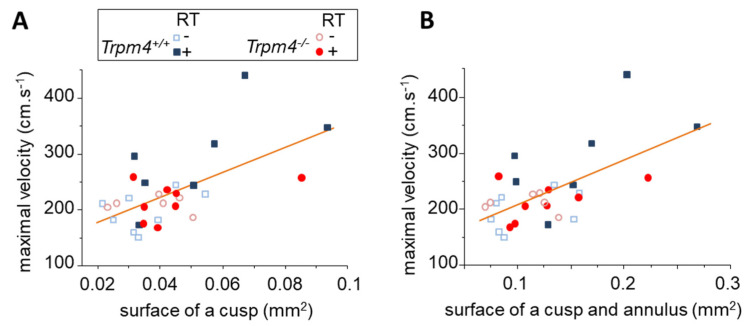
Morphological and functional remodeling relationship. For each mouse, the surface of an aortic valve cusp (**A**) as well as the surface of a cusp and annulus (**B**) were plotted to the maximal aortic valve jet velocity measured at 5 months post-treatment. Each group is represented by a different symbol as indicated in the legend. Altogether, the data were plotted to a linear regression indicating a significant relation between parameters. Pearson’s r was 0.61 for velocity/cusp and 0.62 for velocity/cusp + annulus.

**Figure 5 cancers-14-04477-f005:**
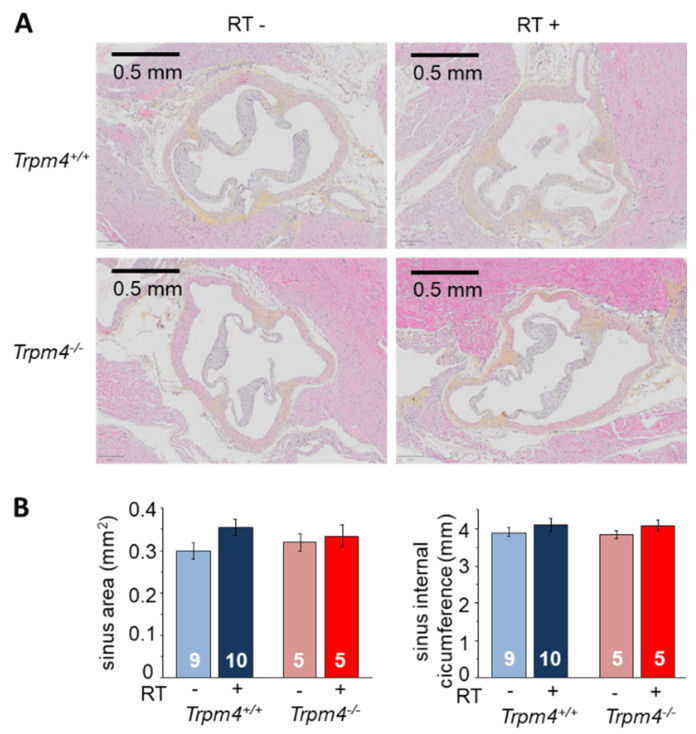
Morphological remodeling after irradiation: sinus remodeling in the transversal axis. (**A**) Histological slice performed at 5 months post-treatment in the transversal axis (3 μm thickness) of the aortic valve representative for each group. Slices were stained with hematoxylin, eosin, and saffron (HES) to reveal cell nuclei (blue) cytoplasm (pink) as well as collagen fibers (yellow). (**B**) The sinus area (**left**) as well as the sinus internal circumference (**right**) was measured for each mouse. Histograms indicate the mean ± S.E.M for each group.

**Figure 6 cancers-14-04477-f006:**
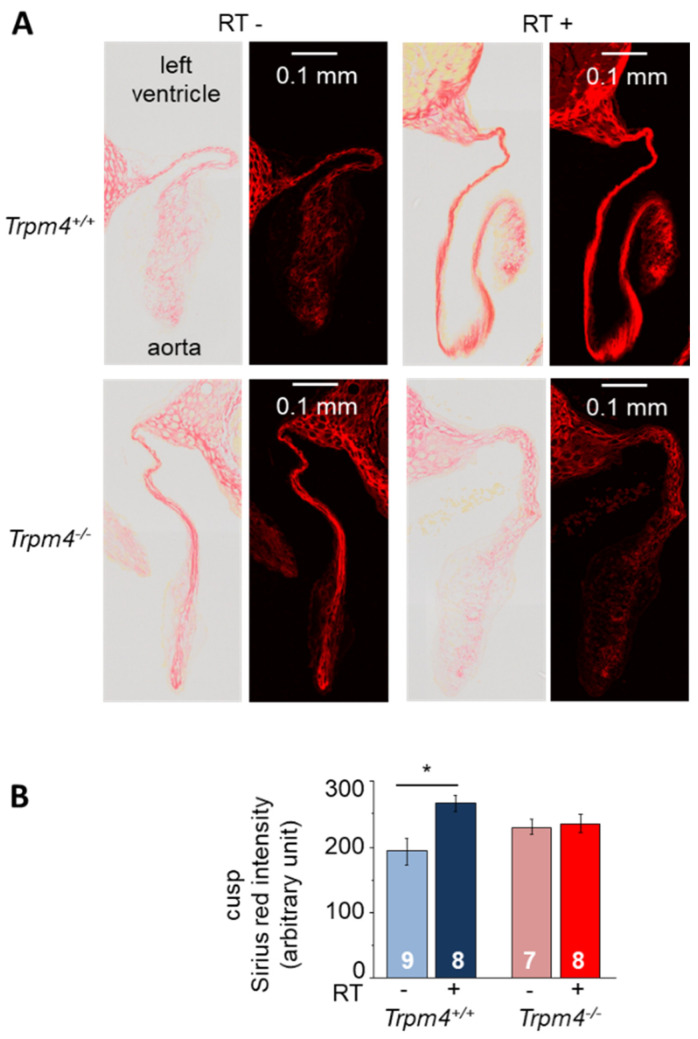
Effect of irradiation on valvular cusp fibrosis. (**A**) Histological slices performed at 5 months post-treatment in the longitudinal axis (3 μm thickness) of the aortic valve representative for each group, stained with Sirius red to reveal fibrosis. For each group, the figure shows the original image with full colors (**left panel**) and the same image with only red chromaticity (see methods) to precisely isolate Sirius red staining. For each panel, the left ventricle is on the upper side and the aorta on the lower side. (**B**) Cusp red chromaticity intensity for each group in arbitrary units (see methods for quantification protocol). Histograms indicate the mean ± S.E.M. for each group. * indicates a significant difference between groups (*p* < 0.05).

**Figure 7 cancers-14-04477-f007:**
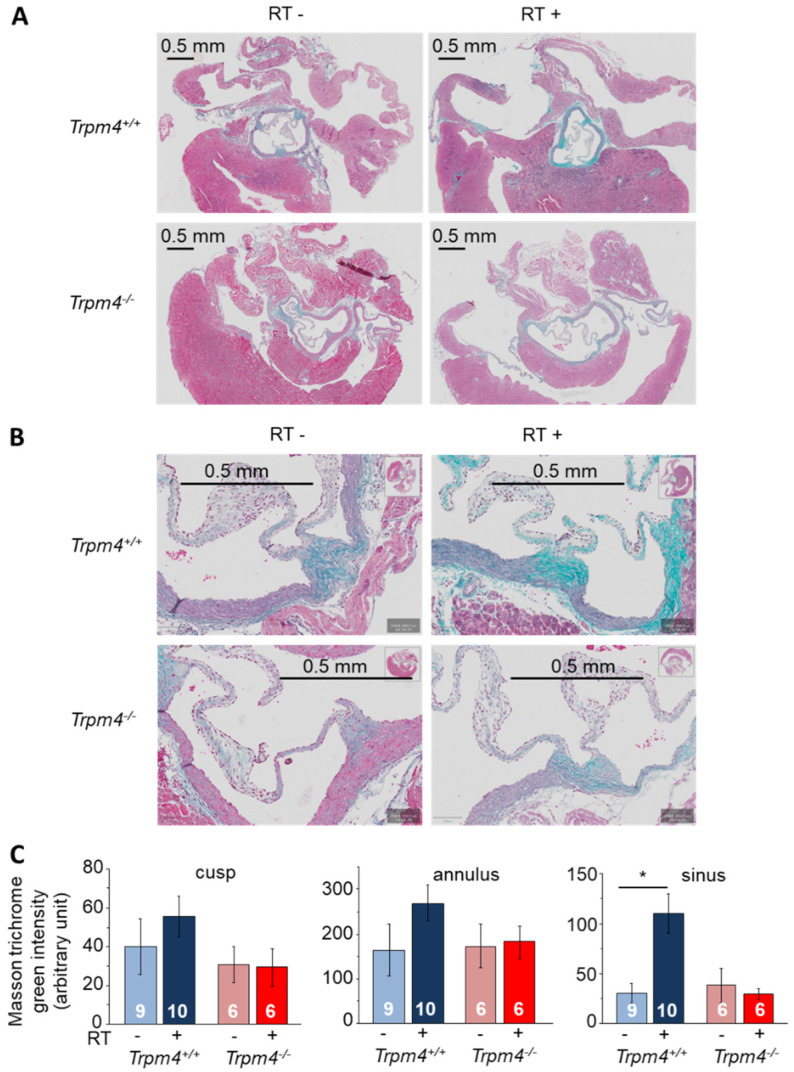
Effect of irradiation on aortic valve fibrosis using Masson trichrome staining. (**A**,**B**) Histological slices performed at 5 months post-treatment in the transversal axis (3 μm thickness) of the aortic valve representative for each group, stained with Masson trichrome to reveal fibrosis. Images in A represent the whole transversal section of hearts while images in B represent a magnification of the same hearts in the aortic valve area. (**C**) Valvular green chromaticity intensity after Masson trichrome staining for each group in arbitrary units (see methods for quantification protocol). Data were measured for the cusp (**left histogram**), annulus (middle histogram), and sinus (**right histogram**). Data indicate the mean ± S.E.M. for each group. * indicates a significant difference between groups (*p* < 0.05).

**Figure 8 cancers-14-04477-f008:**
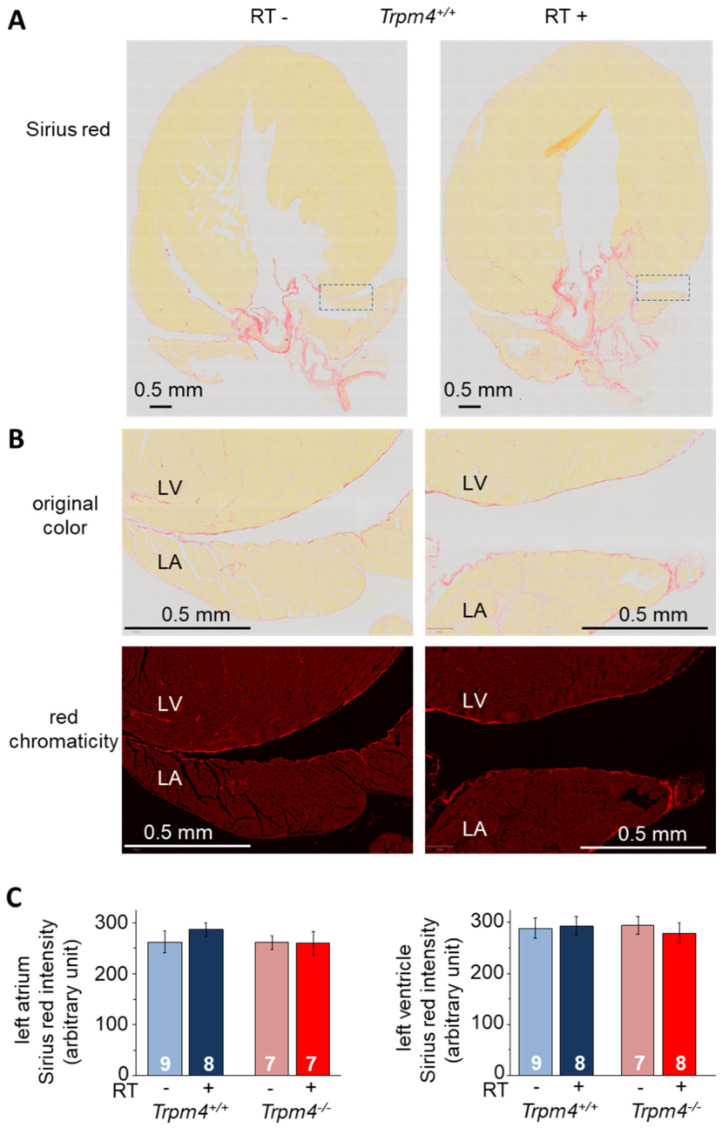
No effect of irradiation on atrial and ventricular fibrosis. (**A**) Histological slices performed at 5 months post-treatment in the longitudinal axis (3 μm thickness) of the whole heart for representative RT− and RT+ Trpm4^+/+^ animals. Slices were stained with Sirius red to reveal fibrosis. (**B**) Magnification of the area indicated in A by dotted lines to reveal the left atrium and ventricle. The upper images are in full color while the bottom images are shown in red chromaticity mode (see methods) to precisely isolate Sirius red staining. For each image, the left ventricle (LV) is on the upper side while the left atrium (LA) is on the lower side. (**C**) The left atrium (**left**) and left ventricle (**right**) red chromaticity intensity after Sirius red staining for each group in arbitrary units (see methods for quantification protocol). Data indicate the mean ± S.E.M. for each group.

**Figure 9 cancers-14-04477-f009:**
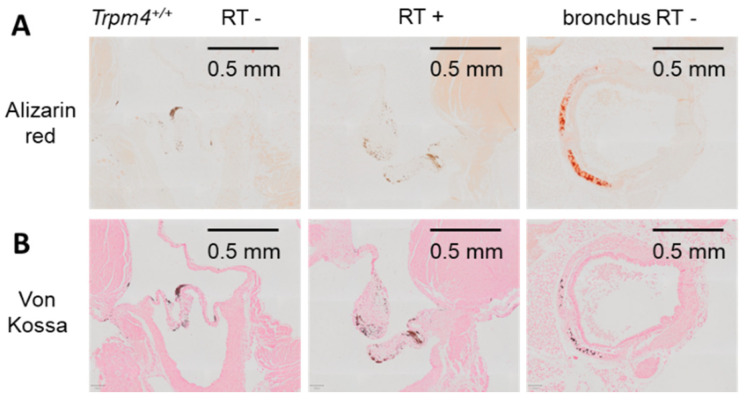
No valvular calcification induced by irradiation. Histological slices performed at 5 months post-treatment in the longitudinal axis (3 μm thickness) of the aortic valve representative for Trpm4^+/+^ mice after radiation treatment or not (**left and middle images**). The right images present a slice which includes a bronchus with cartilage as a control of calcified tissue. (**A**) Slices stained with alizarin red. Red labeling appears in the cartilage from the bronchus but no red staining was detectable on valvular cusps. Note that brown labeling is most likely attributable to melanocytes. (**B**) Slices of the same hearts as in A, stained with Von Kossa.

**Figure 10 cancers-14-04477-f010:**
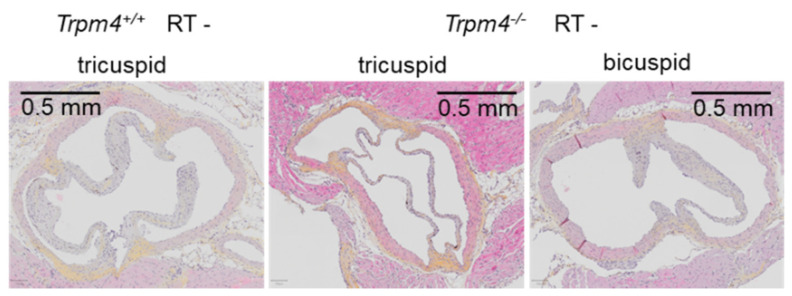
Tricuspid and bicuspid aortic valves in Trpm4^−/−^ mice.

**Table 1 cancers-14-04477-t001:** Echocardiographic findings. Transthoracic echocardiography parameters in 4-month-old mice at baseline (before irradiation) and 5 months after irradiation (M5). Animals were randomly allocated to radiation treatment (RT+) or not (RT−). Data are mean ± S.E.M. * statistically different from +/+ RT−. LA: left atrium; A: aorta; Vmax: maximal velocity; ΔPmean: mean transaortic pressure gradient.

	Trpm4^+/+^ RT−	Trpm4^+/+^ RT+	Trpm4^−/−^ RT−	Trpm4^−/−^ RT+
**Baseline**	*n* = 18	*n* = 20	*n* = 15	*n* = 15
Body weight (g)	28.9 ± 0.5	29.0 ± 0.4	28.9 ± 0.7	28.1 ± 0.7
Heart rate (bpm)	414 ± 12	428 ± 9	400 ± 16	426 ± 16
LA diameter (mm)	1.74 ± 0.07	1.73 ± 0.05	1.78 ± 0.05	1.65 ± 0.07
A root diameter (mm)	1.78 ± 0.03	1.79 ± 0.03	1.73 ± 0.05	1.71 ± 0.05
Aortic valve jet V_max_ (cm.s^−1^)	157.1 ± 7.4	155.7 ± 6.0	158.8 ± 7.4	144.7 ± 4.7
ΔP_mean_ (mmHg)	4.6 ± 0.4	4.6 ± 0.3	5.1 ± 0.5	3.8 ± 0.3
**M5**	*n* = 18	*n* = 20	*n* = 14	*n* = 15
Body weight (g)	33.7 ± 0.6	33.7 ± 0.7	34.2 ± 2.6	33.9 ± 1.1
Heart rate (bpm)	455 ± 12	447 ± 14	430 ± 16	418 ± 16
LA diameter (mm)	1.80 ± 0.05	1.88 ± 0.05	1.80 ± 0.05	1.81 ± 0.04
A root diameter (mm)	1.78 ± 0.04	1.78 ± 0.04	1.89 ± 0.05	1.78 ± 0.04
Aortic valve jet V_max_ (cm.s^−1^)	185.2 ± 7.9	240.9 ± 17.2 *	212.3 ± 11.7	210.9 ± 8.3
ΔP_mean_ (mmHg)	6.5 ± 0.7	11.7 ± 2.1 *	8.3 ± 1.4	8.4 ± 0.6

**Table 2 cancers-14-04477-t002:** Cardiac magnetic resonance imaging findings. CMR imaging parameters in 4-month-old mice at baseline (before irradiation) and 5 months after irradiation treatment (M5). Animals were randomly allocated to radiation treatment (RT+) or not (RT−). Data are mean ± S.E.M. * statistically different from +/+. ^†^ statistically different from +/+ RT−. ^‡^ statistically different from +/+ RT+. LV: left ventricle; LVEDV: left ventricular end diastolic volume; LVESV: left ventricular end systolic volume; LVEF: left ventricular ejection fraction; RVEDV: right ventricular end diastolic volume; RVESV: right ventricular end systolic volume; RVEF: right ventricular ejection fraction.

	Trpm4^+/+^ RT−	Trpm4^+/+^ RT+	Trpm4^−/−^ RT−	Trpm4^−/−^ RT+
**Baseline**	*n* = 17	*n* = 20	*n* = 14	*n* = 15
LV mass/body mass (mg.g^−1^)	2.59 ± 0.11	2.79 ± 0.10	2.92 ± 0.12 *	2.95 ± 0.12 *
LVEDV/body mass (μL.g^−1^)	2.18 ± 0.12	2.25 ± 0.20	1.91 ± 0.16	1.99 ± 0.15
LVESV/body mass (μL.g^−1^)	0.81 ± 0.06	0.97 ± 0.11	0.71 ± 0.08	0.77 ± 0.08
LVEF (%)	63 ± 2	59 ± 3	64 ± 2	62 ± 3
RVEDV/body mass (μL.g^−1^)	0.93 ± 0.08	0.86 ± 0.12	0.97 ± 0.14	0.89 ± 0.10
RVESV/body mass (μL.g^−1^)	0.29 ± 0.04	0.37 ± 0.08	0.30 ± 0.06	0.35 ± 0.09
RVEF (%)	70 ± 3	63 ± 3	70 ± 2	66 ± 4
**M5**	*n* = 18	*n* = 20	*n* = 14	*n* = 15
LV mass/body mass (mg.g^−1^)	2.26 ± 0.14	2.43 ± 0.11	2.77 ± 0.14 ^†^	2.86 ± 0.16 ^‡^
LVEDV/body mass (μL.g^−1^)	1.50 ± 0.18	1.63 ± 0.18	1.60 ± 0.15	1.68 ± 0.18
LVESV/body mass (μL.g^−1^)	0.52 ± 0.09	0.57 ± 0.10	0.62 ± 0.09	0.58 ± 0.10
LVEF (%)	68 ± 3	69 ± 3	62 ± 3	67 ± 3
RVEDV/body mass (μL.g^−1^)	0.36 ± 0.11	0.45 ± 0.09	0.52 ± 0.11	0.62 ± 0.15
RVESV/body mass (μL.g^−1^)	0.14 ± 0.06	0.11 ± 0.02	0.20 ± 0.07	0.16 ± 0.05
RVEF (%)	67 ± 2	72 ± 2	65 ± 4	71 ± 4

## Data Availability

The dataset generated during and/or analyzed during the current study are available from the corresponding author on reasonable request.

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
