# Peer review of "TRPM4 Participates in Irradiation-Induced Aortic Valve Remodeling in Mice"

_cancers, 2022, doi:10.3390/cancers14184477_

Round 1

Reviewer 1 Report

In this article the authors evaluate the role of TRPM4 in the development of aortic valve stenosis following irradiation as after treatment of Hodkin's or breast cancer.

This article is interesting, but I am currently not convinced by the results of this study which are not sufficient. The manuscript in its current state cannot be published. So the authors can hypothesize, but they have to be careful in their conclusions.

Major comments

1) To be sure to irradiate only the aortic valve, the first thing is to do a phospho H2AX marking between 30 min and a few hours after irradiation because of even with all the precautions and the precisions of the new small animal irradiation devices, we will inevitably touch the top of the left ventricle and the atrium. therefore we must know the irradiated part of the heart. 

2) Why did you choose 10 Gy as an acute dose when radiotherapy is done in fractionated form and several doses are delivered to the heart? Why didn't you also use females in the study (as the authors refer to breast cancer in addition to Hodkin's lymphoma)? 

3) the irradiations are carried out under 5% isofluorane it is enormous for a mouse during how long they are left with this concentration? and not at all obligatory especially that the ultrasounds are carried out with 1,5 % and it seems to me that the time of the ultrasound is longer. Especially since isofluorane could have beneficial effects on irradiation (see publication below)

DNA damage assessment in peripheral blood of Swiss albino mice after combined exposure to volatile anesthetics and 1 or 2 Gy radiotherapy in vivo. Vesna Benković et al.; Int J Radiat Biol;. 2021;97(10):1425-1435. doi: 10.1080/09553002.2021.1962565. Epub 2021 Aug 6.

4) figure 1 it would be necessary to add an echo-doppler image allowing to visualize where the measurement is made because it seems that the measurement is taken on the descending aorta but the valve is rather on the ascending side, could you also specify it in the text of the materials and methods

5) I do not agree with the authors' sentence on page 2, line 54, which states that cardiomyocytes are not radiosensitive; many publications state the opposite:

Electrophysiologic and cellular characteristics of cardiomyocytes after X-ray irradiation; Johannes L. Frieß et al.; Mutation Research/Fundamental and Molecular Mechanisms of Mutagenesis; Volume 777, July 2015, Pages 1-10

Jae Sik Kim et al. Impact of High-Dose Irradiation on Human iPSC-Derived Cardiomyocytes Using Multi-Electrode Arrays: Implications for the Antiarrhythmic Effects of Cardiac Radioablation; Int. J. Mol. Sci. 2022, 23, 351.

Virginie Monceau et al.; Epac contributes to cardiac hypertrophy and amyloidosis induced by radiotherapy but not fibrosis; Radiotherapy and Oncology, Volume 111, Issue 1, April 2014, Pages 63-71;

Benjamin V. Becker et al. (2018) Gene expression changes in human iPSCderived cardiomyocytes after X-ray irradiation, International Journal of Radiation Biology, 94:12, 1095-1103, DOI: 10.1080/09553002.2018.1516908

6) on the other hand, TRPM4 is also present in cardiomyocytes, thus with a possible effect on the valve after irradiation: Role of the TRPM4 channel in mitochondrial function, calcium release, and ROS generation in oxidative stress ; Chen Wang et al. Biochemical and Biophysical Research Communications ; Volume 566, 20 August 2021, Pages 190-196;

Genetic background influences expression and function of the cation channel TRPM4 in the mouse heart. Rebekka Medert et al. Basic Res Cardiol . 2020 Nov 17;115(6):70. doi: 10.1007/s00395-020-00831-x.

In table 2 we observe a significantly larger heart mass in the mutant vs wild type but no effect on EF% heart function how would you explain this?

In addition, TRPM4-/- mice have a lower baseline heart rate and is studied in contractility and rhythm disorders so how can you be sure that it is TRPM4 in endothelial cells that is solely responsible for this radiation-induced aortic stenosis? you need to co-label CD31 and TRPM4 on heart sections to be sure of this conclusion

7) in figure 3 we have the impression that the diameter of the aorta is increased in the irradiated mutant and non-mutant mice while in your table 1 there are no differences, how would you explain this?

8) figure 8 put the classic Sirius red version and the yellow and red, it is not possible after 5 months post-IR that no fibrosis is visible with 10Gy and give a percentage

Minor comments

1)      Homogenize the scale bars (with size, 0.5; 1mm) and put one in each figure.

2)      Figure 3 the global image insert with the whole heart is too small and not useful, either remove or enlarge.

3)      When you write HES just hematoxilin and eosin it is not rigorous because you also have saffron and yellow collagen staining (p8 line 259)

4)      Figure 6 A shows as for the irradiated a piece of yellow heart (LV or atrium) and are the staining done at the same time?

Reviewer 2 Report

Dear Editor,

The manuscript submitted by Bangando et al is a morphological and functional study on the effects of radiation on aortic valve remodelling. The study was well performed, but it can be presented better. The main issue namely the association between the target gene (TRPM4) and all pathological events after radiation is very poor.

The study as such failed to justify why TRPM4 was chosen for the investigation.

The study also suffers from the lack of a mechanistic link between the channel and radiation injury.

The authors need to add experiments to improve the observation.

Even though the molecular investigation of this event was not the subject of the study, the authors failed to argue the observations in the best possible way in the discussion. The general consideration of DNA damage or ROS is not even appropriate speculation for the conclusion.

The introduction must be adapted to the aim of the experiment and corrected for some arguments:

The authors argue that the endothelium and fibroblasts are the main targets in the irradiated heart and exclude the cardiomyocytes because of the low sensitivity... a not very correct conclusion that is easily refuted by the mass of publications on the effects of irradiation on energy metabolism, mitochondrial dysfunction and effects on the contraction apparatus in cardiomyocytes. 

Referring to Ref 8 as an example for induction of fibrosis-related proteins after radiation exposure is also not the best.. where there are known studies in radiation research documenting fibrosis and the involvement of molecular signalling (TGF beta, SMAD and MAPK) using omics platforms or classical stains. The authors need to refer to those original studies.

The conclusion drawn between lines 64 and 66 is not sufficient to justify the role of a channel in radiation-induced remodelling in the context of valve damage. The authors need to improve this statement. The authors need to write about the properties of TRPM4 and the signalling pathway it contributes to.

It is also not clear to the reader what is the direct effect of radiation exposure on TRPM4 gene or protein. Is it already known? Has it already been published? How was it affected by irradiation in these cells? Is it upregulated or downregulated?

The authors also need to improve the discussion. Again, speculation and suggestions remain weak. Some questions need to be addressed; How can changes in TRPM4 contribute to the induction of fibrosis? What molecular mechanism is involved in the induction or inhibition of fibrosis?

What is the relationship between TRPM4 and ECM proteins? Is the event consequence of excessive wound healing from a radiation-induced injury to the heart valve?

Does the disruption of Ca +2 homeostasis lead to an increase in growth factors (TGF, CTGF) and initiate the differentiation of fibroblasts? Is fibroblast activation associated with the SMAD or non-SMAD signalling pathway?

The authors need to add experiments to address these issues

I hope addressing these issues improves the manuscript for publication.

Sincerely

Round 2

Reviewer 1 Report

I would like to underline the rigor with which the authors have answered the comments which clearly improves the manuscript. Nevertheless, in the discussion L413- L420, I do not agree with your sentence because many studies show that 5Gy, 2Gy 1Gy or 0, 2 Gy induces long term cardiac fibrosis when the whole heart is irradiated so if you do not have fibrosis in these areas then your system of localized irradiation (at the level of the valve) is really effective but do not say that 5Gy is too low a dose to obtain fibrosis in the long term. 

Author Response

Response to reviewer 1

We sincerely thank the reviewer for his very helpful comments on the first version of the manuscript which strongly help us to improve it.

According to the last comment about the sentences lines 412-420: we agree with the reviewer that our data do not demonstrate a lower sensitivity of ventricular and atrial cells compared to valvular cells. Thus, in the last version, we removed the corresponding sentences. Note that the specific targeting of the valve remains discussed in other parts of the manuscript.

Reviewer 2 Report

Dear Editor,

the authors have addressed my main concerns and comments.
Yours sincerely

Author Response

We sincerely thank the reviewer for his precise reading on the first version of the manuscript and his sharp comments which strongly help us to improve it.